# Theoretical Study on Widening Bandwidth of Piezoelectric Vibration Energy Harvester with Nonlinear Characteristics

**DOI:** 10.3390/mi12111301

**Published:** 2021-10-23

**Authors:** Zhang Qichang, Yang Yang, Wang Wei

**Affiliations:** 1Department of Mechanics, School of Mechanical Engineering, Tianjin University, Tianjin 300072, China; qzhang@tju.edu.cn (Z.Q.); yangyag@tju.edu.cn (Y.Y.); 2Tianjin Key Laboratory of Nonlinear Dynamics and Control, Tianjin University, Tianjin 300072, China

**Keywords:** strong nonlinearity, piezoelectric vibration energy harvester, primary resonance, subharmonic resonance, bandwidth, complex dynamic frequency (CDF) method

## Abstract

In order to make a piezoelectric vibration energy harvester collect more energy on a broader frequency range, nonlinearity is introduced into the system, allowing the harvester to produce multiple steady states and deflecting the frequency response curve. However, the harvester can easily maintain intra-well motion rather than inter-well motion, which seriously affects its efficiency. The aim of this paper is to analyze how to take full advantage of the nonlinear characteristics to widen the bandwidth of the piezoelectric vibration energy harvester and obtain more energy. The influence of the inter-permanent magnet torque on the bending of the piezoelectric cantilever beam is considered in the theoretical modeling. The approximate analytical solutions of the primary and 1/3 subharmonic resonance of the harvester are obtained by using the complex dynamic frequency (CDF) method so as to compare the energy acquisition effect of the primary resonance and subharmonic resonance, determine the generation conditions of subharmonic resonance, and analyze the effect of primary resonance and subharmonic resonance on broadening the bandwidth of the harvester under different external excitations. The results show that the torque significantly affects the equilibrium point and piezoelectric output of the harvester. The effective frequency band of the bistable nonlinear energy harvester is 270% wider than that of the linear harvester, and the 1/3 subharmonic resonance broadens the band another 92% so that the energy harvester can obtain more than 0.1 mW in the frequency range of 18 Hz. Therefore, it is necessary to consider the influence of torque when modeling. The introduction of nonlinearity can broaden the frequency band of the harvester when it is in primary resonance, and the subharmonic resonance can make the harvester obtain more energy in the global frequency range.

## 1. Introduction

The battery is a traditional energy storage device. Many devices (such as intelligent home equipment, implantable medical equipment, environmental monitors) rely on batteries for energy supply. However, the battery cannot achieve long-term energy supply due to limited life, energy storage, and power efficiency and often needs frequent replacement or maintenance. An increasing number of researchers have paid attention to topics such as collecting energy from the environment to eliminate the demand for batteries or extend batteries’ life. Vibration energy is typical mechanical energy in the environment. There are many ways to produce it, such as the surrounding environment, wind flow, fluid movement, mechanical work, and human behavior. Piezoelectric vibration energy harvesters are widely used in research for low power electronic devices such as embedded electronic devices, implantable biomedical devices, wireless sensor nodes, and portable electronic devices due to their simple structure, long life, high power density, and no need for an initial voltage source.

The piezoelectric cantilever beam is the most commonly studied and discussed structure in a piezoelectric vibration energy harvester because of its simple structure and convenient modeling. In the linear piezoelectric cantilever beam structure, the mass is usually added to the free end of the cantilever beam [1], which can reduce and adjust the system’s resonance frequency and improve the mechanical response and output power under low-frequency excitation. However, the main limitation of linear vibration energy harvester is that the natural frequency is single, and the resonance peak is very narrow. When the excitation frequency deviates from the resonant frequency of the harvester, very little power can be generated [2]. The vibration frequency of the environment usually fluctuates in a specific range. In order to obtain better output performance and maintain higher acquisition efficiency, a broadband harvester is required. Introducing nonlinearity into the harvester can broaden the frequency bandwidth of the system, increase the mechanical response amplitude and power output. Nonlinear spread spectrum technology makes use of nonlinear stiffness [3,4], the piecewise linear effect generated by the collision process [5,6], and multi-steady-state motion [7,8] to deflect the amplitude–frequency curve of the energy harvester, ensure a large amplitude in a wide frequency range, achieve the effect of broadening the frequency band, and improve the acquisition efficiency of the energy harvester.

The realization of the multi-steady state of the piezoelectric vibration energy harvester mainly uses the magnetic field force provided by permanent magnets. A different number of steady states can be obtained by using different arrangement modes of permanent magnets. Erturk et al. [9] propose a bistable piezoelectric vibration energy harvester with two stable positions. The device consists of a piezoelectric cantilever beam and three permanent magnets. A permanent magnet is installed at the tip of the cantilever beam, and two permanent magnets are fixed near the tip of the cantilever beam. The experimental results show that compared with the linear harvester, the system has a wider effective frequency band and higher output power. Since then, bistable piezoelectric vibration energy harvesters have been studied by many researchers. Zhou et al. [10] propose a triple steady-state piezoelectric vibration energy harvester, which consists of a piezoelectric cantilever beam with a permanent magnet at the tip and two external permanent magnets. Compared with the bistable energy harvester with deeper potential wells, the triple steady-state arrangement has a shallow potential well, which allows the harvester to easily produce large amplitude vibration between wells and higher energy output on a broader frequency range.

For the nonlinear piezoelectric energy harvester, the forward sweep can obtain a wider effective bandwidth than the reverse sweep when the sweep is conducted. Therefore, the nonlinear harvester cannot always obtain a higher amplitude as the external excitation frequency varies. Qing et al. [11] propose a lambda-shaped piezoelectric energy harvester, which can produce much higher power than the cantilever piezoelectric energy harvester but still has the shortcomings of the linear energy harvester. Li et al. propose [12] a generalized multi-mode piezoelectric energy harvester. The model consists of the main cantilever beam and multiple branches. The effective bandwidth of the harvester can be adjusted by changing the branch parameters and tip mass. It can overcome the drawbacks of the linear piezoelectric energy harvester and avoid the drawbacks mentioned above of nonlinear energy harvesters. Sun et al. [13] propose a horizontal asymmetric U-shaped piezoelectric energy harvester model by combining multi-mode and nonlinear force. The asymmetric U-shaped structure gives the model multi-mode properties, while the nonlinearity introduced by magnetic force improves the energy output of the harvester, reduces the resonant frequency, and broadens the effective bandwidth. The above studies improved the efficiency of energy acquisition by changing the structure of the piezoelectric energy harvester. However, the complex structure may have problems in practical application and model manufacturing.

The frequency sweep of a nonlinear piezoelectric energy harvester cannot reveal all the steady-state behaviors it can achieve, and frequency sweep does not reveal the subharmonic motion. Syta et al. [14] studied the subharmonic vibration of a bistable harvester at a specific frequency, and the system showed different motion behaviors under different initial conditions. Huguet et al. [15,16,17] propose a complete subharmonic orbit analysis to predict the contribution to the global bandwidth of bistable energy harvesters and study the effects of different parameters on the response through experiments. Syta and Huguet mainly studied subharmonic motion by numerical simulation and experimental methods but did not study the influence of subharmonics on broadening the frequency band under different external excitations. Huguet also used the harmonic balance method to study the robustness of subharmonic motion, but for strongly nonlinear systems, sufficient harmonic terms are needed to obtain more accurate asymptotic solutions. Therefore, this paper theoretically investigates how to make full use of nonlinear characteristics to broaden the bandwidth of the energy harvester of the piezoelectric cantilever beam structure.

The theoretical method for solving strongly nonlinear vibration has always been a difficulty. In order to solve this problem, Wang et al. [18] proposed a single-degree-of-freedom complex dynamic frequency (CDF) method, applied it to the study of strongly nonlinear vibration energy harvester, and experimentally verified the method’s effectiveness. This paper extends this method based on this theory and studies the above strongly nonlinear vibration problem.

In modeling the piezoelectric cantilever beam harvester, the magnetic repulsion between permanent magnets is generally considered [19,20]. The torque between permanent magnets also affects the bending of piezoelectric cantilever beams, thereby affecting the piezoelectric effect. Kim et al. [21] established a model considering the torque between permanent magnets but did not analyze the influence of torque. This paper discusses the influence of torque on the piezoelectric cantilever vibration energy harvester.

In this paper, the energy harvester of the piezoelectric cantilever beam structure is studied. The permanent magnets provided nonlinearity for the harvester, and the magnetic repulsion and torque between them are considered. In order to achieve a bistable state, the distance between the free-end permanent magnet of the cantilever beam and the fixed permanent magnet is determined according to the static analysis. The approximate analytical solution and amplitude–frequency relationship of the strongly nonlinear harvester’s steady-state motion are obtained using the CDF method. The broadening effect of primary resonance and subharmonic resonance on the bandwidth of the harvester is compared and analyzed.

The rest of this paper is organized as follows. Section 2 gives the piezoelectric cantilever beam dynamics model considering the torque between permanent magnets. Section 3 outlines the static analysis of the system. In Section 4, the approximate analytical solutions and amplitude–frequency relations of the system’s primary and subharmonic resonance are obtained using the CDF method. Section 5 analyzes the system’s dynamic characteristics and compares the differences in the energy obtained by the primary resonance and the 1/3 subharmonic resonance in the global operating frequency range and the variation of the subharmonic response with different external excitations. Section 6 summarizes the full-text results.

## 2. Theoretical Model

The piezoelectric material as an energy transfer device directly affects the collection efficiency of the harvester. Currently, commonly used piezoelectric materials [22,23,24] include piezoelectric ceramics, piezoelectric polymer, and piezoelectric composites. Piezoelectric ceramics have a high electromechanical coupling coefficient and low cost but are brittle and easily fractured, piezoelectric polymers are flexible but have low electromechanical coupling coefficients, and piezoelectric composites combine the advantages of piezoelectric ceramics and piezoelectric polymers with excellent piezoelectric properties and strong flexibility. Therefore, in this paper, piezoelectric composites are used as piezoelectric materials of the piezoelectric energy harvester.

The mechanical model of the piezoelectric cantilever beam is shown in Figure 1. *L* is the length of the composite beam, which consists of a substrate beam and two piezoelectric plates. The materials of the substrate and piezoelectric plate are beryllium bronze QBe1.9 and FMC M2807 P2. A layer of the piezoelectric plate is pasted on the substrate’s top and bottom surfaces, and the substrate is rigidly connected to the piezoelectric plates without considering the influence of the multilayer beam. One end of the composite beam is fixed on base C, and the other is free. Permanent magnet A is fixed to the free end of the composite beam, and permanent magnet B is fixed to another side of base C. The two permanent magnets are mutually exclusive and horizontally aligned. When the system is in equilibrium, the gravity of the composite beam and permanent magnet A has no effect on the deformation of the piezoelectric cantilever beam. *P*(*t*) is the basic motion; *x* is the axial spatial coordinate; *t* is the time coordinate; *z*(*x*,*t*) is the transverse deformation of the position on the cantilever beam at any time; and *F_x_*, *F_z_*, and *M_c_* are the magnetic force in the *x*-direction, the magnetic force in the *z*-direction, and the torque in the *y*-direction of the permanent magnet A under the action of the permanent magnet B, respectively. The physical parameters and material properties of the piezoelectric cantilever beam vibration system are shown in Table 1.

Due to the sizeable length/thickness ratio of the beam, this paper only considers the bending deformation of the piezoelectric cantilever beam; regardless of the shear deformation and the influence of the rotational inertia of the section around the neutral axis, the piezoelectric cantilever beam is regarded as an Euler–Bernoulli beam. The nonlinear magnetic force at the free-end boundary is regarded as a concentrated load, as shown in Figure 2.

The influence of temperature and residual stress on the piezoelectric cantilever beam is not considered in the modeling. According to the Hamilton principle, the governing equation of the harvester can be written as
(1)$$\delta \int_{t_{1}}^{t_{2}}\left(T-U\right)dt+\delta \int_{t_{1}}^{t_{2}}Wdt=0,$$
where *T* is the system’s kinetic energy, *U* is the system’s potential energy, *W* is the energy of the magnetic repulsion and torque, and δ is the variational symbol. According to Figure 1, the kinetic energy of the substrate is
(2)$$T_{s}=\frac{1}{2}\rho_{s}A_{s}\int_{0}^{L}{\left[\dot{z}(x,t)+\dot{P}(t)\right]}^{2}dx,$$
where $A_{s}$ is the section area of the substrate, and $\rho_{s}$ is the substrate’s material density. The kinetic energy of the piezoelectric plate is
(3)$$T_{p}=\rho_{p}A_{p}\int_{0}^{L_{p}}{\left[\dot{z}(x,t)+\dot{P}(t)\right]}^{2}dx,$$
where $A_{p}$ is the section area of the piezoelectric plate, and $\rho_{p}$ is the piezoelectric plate’s material density. The kinetic energy of free-end permanent magnet A is
(4)$$T_{m}=\frac{1}{2}m_{a}{\left[\dot{z}\left(L,t\right)+\dot{P}\left(t\right)\right]}^{2}+\frac{1}{2}J_{a}{\left[{\frac{\partial z\left(x,t\right)}{\partial x\partial t}|}_{x=L}\right]}^{2},$$
where $m_{a}$ is the mass of permanent magnet A, and $J_{a}$ is the rotational inertia of permanent magnet A. The bending strain energy of the substrate is
(5)$$U_{s}=\frac{1}{2}E_{s}I_{s}\int_{0}^{L}{\left[z^{\prime \prime }(x,t)\right]}^{2}dx,$$
where $E_{s}$ is the substrate’s Young’ s modulus, and $I_{s}$ is the section inertia moment of the substrate, $I_{s}=bh_{s}^{3}/12$. The bending strain energy of the piezoelectric plate is
(6)$$H_{p}=E_{p}I_{p}\int_{0}^{L_{p}}{\left[z^{\prime \prime }(x,t)\right]}^{2}dx-\frac{1}{2}e_{31}b(h_{p}+h_{s})\dot{\lambda }(t)z^{\prime }(L_{p},t)-\frac{1}{4}C_{p}\dot{\lambda }{\left(t\right)}^{2},$$
where $E_{p}$ is the piezoelectric plate’s Young’ s modulus, $I_{p}=(4h_{p}^{2}+6h_{p}h_{s}+3h_{s}^{2})bh_{p}/12$ is the inertia moment of the piezoelectric plate, $e_{31}=E_{p}d_{31}$ is the effective piezoelectric stress constant, *b* is the width of the substrate, $h_{s}$ and $h_{p}$ are the substrate and piezoelectric thickness, λ(t) is the flux linkage, $C_{p}=b\varepsilon_{33}^{S}L_{p}/h_{p}$ is the capacitance, $\varepsilon_{33}^{S}=\varepsilon_{31}\varepsilon_{0}$ is the dielectric constant of medium, and $\varepsilon_{31}$ is the relative dielectric constant of the piezoelectric plate.

By considering the first-order vibration mode of the piezoelectric cantilever beam, we can set
(7)z(x,t)=η(t)ϕ(x),
where ϕ(x) is the vibration mode function of the beam, and η(t) indicates the vibration mode of the beam as a function of the generalized time coordinate. Since the research object is a variable-section beam, the vibration mode function of the beam needs to be expressed in segments,
(8)$$\phi (x)=\left\{ \begin{array}{l} \phi_{1}(x),0<x<L_{p},\\ \phi_{2}(x),L_{p}<x<L, \end{array}\right.,$$
(9)$$\phi_{i}(x)=c_{i1}\sin\beta_{i}+c_{i2}\cos\beta_{i}+c_{i3}\sinh\beta_{i}+c_{i4}\cosh\beta_{i}$$
where $c_{i1}\sim c_{i4}(i=1,\;2)$ are determined by the piezoelectric cantilever beam boundary conditions and continuity conditions, and $\beta_{i}$ represents eigenvalues.

Let the virtual work of moment $M_{y0}$, $M_{yL}$ and shear $Q_{z0}$, $Q_{zL}$ acting on the ends of the cantilever beam at the corresponding virtual displacement on the boundary be
(10)$$\int_{t_{1}}^{t_{2}}\left[Q_{z0}\delta z\left(0,t\right)-Q_{zL}\delta z\left(L,t\right)-M_{y0}\delta z^{\prime }\left(0,t\right)+M_{yL}\delta z^{\prime }\left(L,t\right)+I\left(t\right)\delta \lambda \right]dt,$$
where $\delta z(x,t)=\delta \sum_{i=1}^{n}\eta_{i}(t)\phi_{i}(x)=\sum_{i=1}^{n}\phi_{i}(x)\delta \eta_{i}(t)$, $I(t)=-\dot{\lambda }(t)/R_{L}$. Equations (2)–(10) are substituted into Equation (1) and organized to obtain the system control equation
(11)$$\ddot{\eta }(t)+2\xi \omega \dot{\eta }(t)+\omega^{2}\eta (t)+\theta v(t)=-\gamma \ddot{P}(t)-Q_{zL}\phi_{2}(L)+M_{yL}\phi_{2}{}^{\prime }(L),$$
(12)$$\frac{1}{2}C_{p}\dot{v}(t)-\theta \dot{\eta }(t)+\frac{v(t)}{R_{L}}=0$$
where ξ is the damping ratio. The damping source in the system consists of mechanical damping and air damping, and the specific value of the damping ratio is determined by the experiment; ξ=0.0178 is taken in this paper. ω is the natural frequency of the system, $v(t)=-\dot{\lambda }(t)$ is the voltage, $\gamma =(\rho_{s}A_{s}+2\rho_{p}A_{p})\int_{0}^{L_{p}}\phi_{1}(x)dx+\rho_{s}A_{s}\int_{L_{p}}^{L}\phi_{2}(x)dx+m_{a}\phi_{2}(L)$ is the base excitation coefficient, and $\theta =e_{31}b(h_{p}+h_{s})\phi_{1}{}^{\prime }(L_{p})/2$ is the electromechanical coupling terms. The normalization condition is
(13)$$(\rho_{s}A_{s}+2\rho_{p}A_{p})\int_{0}^{L_{p}}\phi_{1}{\left(x\right)}^{2}dx+\rho_{s}A_{s}\int_{L_{p}}^{L}\phi_{2}{\left(x\right)}^{2}dx+m_{a}\phi_{2}{\left(L\right)}^{2}+J_{a}\phi_{2}{}^{\prime }{\left(x\right)}^{2}=1,$$
(14)$$(E_{s}I_{s}+2E_{p}I_{p})\int_{0}^{L_{p}}\phi_{1}{}^{\prime \prime }{\left(x\right)}^{2}dx+E_{s}I_{s}\int_{L_{p}}^{L}\phi_{2}{}^{\prime \prime }{\left(x\right)}^{2}dx=\omega^{2},$$

In order to obtain the natural frequency and mode of the system, the Euler–Bernoulli beam transverse vibration equation without damping free vibration [25] is used,
(15)$$EI(x)\frac{\partial^{4}z(x,t)}{\partial x^{4}}+m(x)\frac{\partial^{2}z(x,t)}{\partial t^{2}}=0,$$
where EI(x) is flexural stiffness of composite section, and m(x) is mass per unit length of composite beam. Substituting Equation (7) into Equation (15) yields
(16)$$EI(x)\frac{d^{4}\phi (x)}{dx^{4}}-m(x)\omega^{2}\phi (x)=0,$$
where ω is the natural frequency of the system. Equation (16) can be rewritten as
(17)$$\frac{d^{4}\phi (x)}{dx^{4}}-\beta^{4}\phi (x)=0,$$
where $\beta^{4}=m(x)\omega^{2}/EI(x)$. For the segmented beam whose vibration mode function is shown in Equation (8), we have
(18)$$\beta_{1}{}^{4}=\frac{(\rho_{s}A_{s}+2\rho_{p}A_{p})\omega^{2}}{E_{s}I_{s}+2E_{p}I_{p}},\;\beta_{2}{}^{4}=\frac{\rho_{s}A_{s}\omega^{2}}{E_{s}I_{s}},$$
fixed-end boundary conditions:(19)$$\left\{ \begin{array}{l} {\phi_{1}(x)|}_{x=0}=0,\\ {\phi_{1}{}^{\prime }(x)|}_{x=0}=0, \end{array}\right.$$
free-end boundary conditions:(20)$$\left\{ \begin{array}{l} {\left[E_{s}I_{s}\phi_{2}{}^{\prime \prime }(x)-\omega^{2}J_{a}\phi_{2}{}^{\prime }(x)\right]}_{x=L}=0,\\ {\left[E_{s}I_{s}\phi_{2}{}^{\prime \prime \prime }(x)+\omega^{2}m_{a}\phi_{2}(x)\right]}_{x=L}=0, \end{array}\right.$$
and continuity conditions:(21)$$\left\{ \begin{array}{l} \phi_{1}(L_{p})=\phi_{2}(L_{p}),\\ {\phi_{1}{}^{\prime }(x)|}_{x=L_{p}}={\phi_{2}{}^{\prime }(x)|}_{x=L_{p}},\\ {\left(E_{s}I_{s}+2E_{p}I_{p})\phi_{1}{}^{\prime \prime }(x\right)|}_{x=L_{p}}=E_{s}I_{s}{\phi_{2}{}^{\prime \prime }(x)|}_{x=L_{p}},\\ {\left(E_{s}I_{s}+2E_{p}I_{p})\phi_{1}{}^{\prime \prime \prime }(x\right)|}_{x=L_{p}}=E_{s}I_{s}{\phi_{2}{}^{\prime \prime \prime }(x)|}_{x=L_{p}}, \end{array}\right.$$

By combining the eight Equations in (19)~(21), we can obtain the linear homogeneous equations for $c_{11}\sim c_{14}$ and $c_{21}\sim c_{24}$. In order to obtain the non-zero solution, the determinant of the coefficient matrix is set to zero, and the first-order natural frequency of the system can be obtained as *ω* = 154 rad/s. By using the above normalization conditions, all the unknown variables $c_{11}\sim c_{14}$ and $c_{21}\sim c_{24}$ can be obtained, and the modal function of the piezoelectric cantilever beam is
(22)$$\left\{ \begin{array}{l} \phi_{1}(x)=7.6745\sin(13.65x)-8.4921\cos(13.65x)\\ \;\;-7.6745\sinh(13.65x)+8.4921\cosh(13.65x)\\ \phi_{2}(x)=8.89501\sin(14.99x)-13.8796\cos(14.99x)\\ \;\;-15.4554\sinh(14.99x)+15.5211\cosh(14.99x) \end{array}\right..$$

## 3. Static Analysis

In the study of nonlinear vibration energy harvesters, many researchers [26,27,28] provide nonlinear force for the vibration energy harvester by introducing magnetic force. The magnetic dipole method and the magnetization current method are common methods to calculate magnetic force. Tan et al. [29] prove that both methods produce errors when the permanent magnet spacing is sufficiently small. However, the magnetic dipole method results in closer calculations to experimental measurements than the magnetization current method. This section firstly uses the magnetic dipole method to derive the magnetic force and torque expressions [30]. Secondly, the equilibrium point of the harvester is determined. Finally, the effect of torque on the harvester is analyzed.

### 3.1. Magnetic Force and Torque

The structure of the piezoelectric cantilever beam is shown in Figure 1, where the permanent magnets A and B are simplified as magnetic dipoles A and B, respectively. The magnetic induction intensity generated by dipole B at the location of dipole A is given by
(23)$$B_{BA}=-\frac{\mu_{0}}{4\pi }\nabla \frac{m_{B}\cdot r}{r^{3}},$$
where $\mu_{0}$ is the vacuum permeability. $m_{B}$ is the magnetic moment of magnetic dipole B, and the size of the magnetic moment is related to the volume of the permanent magnet. For example, $m_{B}=M_{B}V_{B}$, $M_{B}$ is the magnetization of the permanent magnet B, and $V_{B}$ is the volume of the permanent magnet B. For permanent magnets, magnetization can be estimated by residual flux density, $M_{B}=B_{r}/\mu_{0}$. r is the vector from the center of magnetic dipole B to the center of magnetic dipole A. The force and torque applied by magnetic dipole B to magnetic dipole A are respectively
(24)$$F_{BA}=-\nabla \left(-B_{BA}\cdot m_{A}\right)=-\frac{\mu_{0}}{4\pi }\nabla \left[\left(\nabla \frac{m_{B}\cdot r}{r^{3}}\right)\cdot m_{A}\right],$$
(25)$$M_{c}=m_{B}\times B_{\mathrm{BA}}$$
where $m_{A}$ is the magnetic moment of the magnetic dipole A. According to the vector differential principle, the following can be obtained
(26)$$F_{BA}=\frac{3\mu_{0}m_{A}m_{B}}{4\pi r^{4}}\left[\begin{array}{l} \hat{r}\cdot \left({\hat{m}}_{B}\cdot {\hat{m}}_{A}\right)+{\hat{m}}_{A}\left(\hat{r}\cdot {\hat{m}}_{B}\right)\\ +{\hat{m}}_{B}\left(\hat{r}\cdot {\hat{m}}_{A}\right)-5\hat{r}\left(\hat{r}\cdot {\hat{m}}_{A}\right)\left(\hat{r}\cdot {\hat{m}}_{B}\right) \end{array}\right],$$
(27)$$M_{c}=\frac{\mu_{0}m_{A}m_{B}}{4\pi r^{3}}\left[3({\hat{m}}_{B}\cdot \hat{r})({\hat{m}}_{A}\times \hat{r})+({\hat{m}}_{B}\times {\hat{m}}_{A})\right]$$
where $m_{A}=m_{A}{\hat{m}}_{A}$, $m_{B}=m_{B}{\hat{m}}_{B}$ and $r=r\hat{r}$. ${\hat{m}}_{A}$, ${\hat{m}}_{B}$, and $\hat{r}$ are unit vectors.

Figure 3 shows the geometric relationship diagram when the free-end displacement of the piezoelectric cantilever beam is *z*. The magnetic force in the *z*-direction and the torque in the *y*-direction can be obtained as follows
(28)$$F_{z}=\frac{3\mu_{0}m_{A}m_{B}z}{4\pi r^{5}L}\left[d-\sqrt{L^{2}-z^{2}}-\frac{5d}{r^{2}}\left(z^{2}-d\sqrt{L^{2}-z^{2}}\right)\right],$$
(29)$$M_{c}=\frac{\mu_{0}m_{A}m_{B}z}{4\pi r^{5}L}\left(3d^{2}+3d\sqrt{L^{2}-z^{2}}-r^{2}\right).$$

### 3.2. Equilibrium Points of the System

By analyzing the equilibrium point and stability of the autonomous system corresponding to Equations (11) and (12), the static bifurcation characteristics of the piezoelectric cantilever beam system are obtained. Fourier expansion omits higher-order terms for magnetic force and torque applied to the free end of a piezoelectric cantilever beam
(30)$$Q_{zL}=-F_{z}=-\left(a_{1}z+a_{2}z^{3}\right),$$
(31)$$M_{yL}=M_{c}=b_{1}z+b_{2}z^{3},$$
where $a_{1}$, $a_{2}$, $b_{1}$, and $b_{2}$ are the corresponding Fourier expansion coefficients, respectively.

Letting $z_{1}=\eta$, $z_{2}=\dot{\eta }$, and $z_{3}=v$ in Equations (11) and (12), we obtain
(32)$$\left[\begin{array}{l} {\dot{z}}_{1}\\ {\dot{z}}_{2}\\ {\dot{z}}_{3} \end{array}\right]=\left[\begin{array}{c} z_{2}\\ -2\xi \omega z_{2}-\omega^{2}z_{1}+\alpha_{1}z_{1}-\alpha_{2}z_{1}^{3}+\theta z_{3}\\ -\mu z_{3}-\vartheta z_{2} \end{array}\right],$$
where $\alpha_{1}=a_{1}\phi {\left(L\right)}^{2}+b_{1}\phi (L)\phi^{\prime }(L)$, $\alpha_{2}=-a_{2}\phi {\left(L\right)}^{4}-b_{2}\phi {\left(L\right)}^{3}\phi^{\prime }(L)$, $\mu =2/(C_{p}R_{L})$, and $\vartheta =2\theta /C_{p}$.

The system has three fixed points: (0,0,0), $\left(\pm \sqrt{(\alpha_{1}-\omega^{2})/\alpha_{2}},0,0\right)$. According to the Routh–Hurwitz criterion, there is a stable zero fixed point when $\alpha_{1}<\omega^{2}$, and there are two stable nonzero fixed points and an unstable zero fixed point when $\alpha_{1}>\omega^{2}$, so $\alpha_{1}=\omega^{2}$ is a bifurcation point. By substituting the parameters, it can be determined that when the distance between permanent magnets *d* < 25 mm, the system produces bistable motion.

If the term containing damping does not appear in Equation (32), the energy function of the system is
(33)$$L(z_{1},z_{2})=\frac{\omega^{2}-\alpha_{1}}{2}z_{1}^{2}+\frac{\alpha_{2}}{4}z_{1}^{4}+\frac{1}{2}z_{2}^{2},$$

The contour diagram of the energy function $L(z_{1},z_{2})$ shown in Figure 4 is obtained by assigning different initial values to $z_{1}$ and $z_{2}$. There are four different types of curves (or points) in the figure: $L(z_{1},z_{2})>0$, $L(z_{1},z_{2})=0$, $-{\left(\alpha_{1}-\omega^{2}\right)}^{2}/(4\alpha_{2})<L(z_{1},z_{2})<0$, and $L(z_{1},z_{2})=-{\left(\alpha_{1}-\omega^{2}\right)}^{2}/(4\alpha_{2})$ from outside to inside. It can be seen that different initial conditions may lead to different motion modes.

### 3.3. Effect of Torque

Figure 5 shows the static bifurcation diagram of the equilibrium point. It can be seen from the figure that when the distance between the permanent magnets *d* is small, there is almost no difference in the equilibrium position in both cases. When *d* > 20 mm, the effect of torque on the equilibrium position is greater as *d* increases. In order to study the influence of torque on the energy output of the harvester, the effect of torque on the RMS voltage and power is analyzed below when *d* > 20 mm.

Figure 6a gives the RMS voltage versus load resistance for the harvester. The RMS voltage increases as the load resistance increases and eventually increases to saturation. Figure 6b shows the corresponding power versus load resistance, where the power first increases and then decreases with the increasing load resistance. It can be observed from Figure 6 that torque affects voltage and power. Figure 7 shows the relationship between the relative error of the output power and the load resistance without considering the torque. The results show that when the load resistance is small and the distance between permanent magnets is large, not considering that the torque causes a large error, and the maximum error can reach 18.96% when *d* = 24 mm.

## 4. Approximate Analytical Solution

Since the system is a strongly nonlinear vibration system, the approximate analytical solution of the system is obtained using the complex dynamic frequency (CDF) method and compared with the multiscale method and the numerical solution.

For convenience, we rewrite Equations (11) and (12) as
(34)$$\ddot{\eta }(t)+\omega^{2}\eta (t)=f(\eta ,\dot{\eta },v),$$
(35)$$\dot{v}(t)+\mu v(t)-\vartheta \dot{\eta }(t)=0,$$
where $f(\eta ,\dot{\eta },v)=-2\xi \omega \dot{\eta }(t)+\alpha_{1}\eta (t)-\alpha_{2}\eta {\left(t\right)}^{3}-\theta v(t)+\gamma \ddot{P}(t)$, and the expressions of $\alpha_{1}$, $\alpha_{2}$, μ, and ϑ are the same as (32), $\ddot{P}(t)=P\cos(\Omega t+\phi )$.

### 4.1. Complex Dynamic Frequency (CDF) Method of Primary Resonance

The complex dynamic frequency (CDF) method is used to solve the primary resonance solution of the system Equations (34) and (35). It can clearly be seen from Equation (35) that v and η have the same form of analytic expression, so set
(36)$$\left\{ \begin{array}{l} \eta =\zeta +\bar{\zeta },\\ \dot{\eta }=i(\omega_{10}+\varepsilon \omega_{11})(\zeta +\bar{\zeta }), \end{array}\right.$$
(37)$$v=(\Gamma_{1}+i\Gamma_{2})\zeta +(\Gamma_{1}-i\Gamma_{2})\bar{\zeta },$$
where $\zeta =ae^{i\omega_{10}t}/2$, $\bar{\zeta }=ae^{-i\omega_{10}t}/2$, *a* is the amplitude; $\omega_{10}$ is a constant for the undetermined frequency; $\omega_{11}$ is a dynamic frequency, a function of time t; $\Gamma_{1}$ and $\Gamma_{2}$ are undetermined coefficients; and ε is the bookkeeping parameter. The approximate analytical solution of the system can be obtained by applying the CDF method and the harmonic balance method to Equations (34) and (35), respectively,
(38)$$\left\{ \begin{array}{l} \eta =a\cos(\omega_{10}t),\\ \dot{\eta }=-a(\omega_{10}+\omega_{11})\sin(\omega_{10}t), \end{array}\right.$$
(39)$$v=\frac{a\vartheta \omega_{10}}{\mu^{2}+\omega_{10}^{2}}\left[\omega_{10}\cos(\omega_{10}t)-\mu \sin(\omega_{10}t)\right],$$
where $\omega_{11}=a^{2}\alpha_{2}\cos(2\omega_{10}t)/(8\omega_{10})$. Combining Equation (7), we can obtain the actual vibration response,
(40)$$\left\{ \begin{array}{l} z(L,t)=\phi_{2}(L)a\cos(\omega_{10}t),\\ z^{\prime }(L,t)=-\phi_{2}(L)a(\omega_{10}+\omega_{11})\sin(\omega_{10}t), \end{array}\right.$$
and obtain the amplitude–frequency response relation,
(41)$${\left[\frac{4a\left(\omega^{2}-\omega_{10}^{2}\right)-4a\alpha_{1}+3a^{3}\alpha_{2}+4a\Gamma_{1}\theta }{4P\gamma }\right]}^{2}+{\left(\frac{a\nu \omega_{10}+a\Gamma_{2}\theta }{P\gamma }\right)}^{2}=1,$$
(42)$$\tan\phi =\frac{4\left(a\nu \omega_{10}+a\Gamma_{2}\theta \right)}{4a\omega^{2}-4a\omega_{10}^{2}+3a^{3}\alpha_{2}-4\alpha_{1}a+4a\Gamma_{1}\theta },$$
where $\Gamma_{1}=\vartheta \omega_{10}^{2}/(\mu^{2}+\omega_{10}^{2})$, $\Gamma_{2}=\vartheta \mu \omega_{10}/(\mu^{2}+\omega_{10}^{2})$.

Rewriting Equation (39) as
(43)$$v=V\cos(\omega_{10}t+\varphi ),$$
where $V=a\omega_{10}\vartheta /\sqrt{\mu^{2}+\omega_{10}^{2}}$ is the amplitude of voltage, $\cos\varphi =\omega_{10}/\sqrt{\mu^{2}+\omega_{10}^{2}}$, $\sin\varphi =\mu /\sqrt{\mu^{2}+\omega_{10}^{2}}$, the average power is
(44)$$P_{av}=\frac{V^{2}}{2R_{L}},$$

Substituting Equation (44) into (41), and considering $f=\omega_{10}/(2\pi )$, we can obtain the relationship between average power $P_{av}$ and excitation frequency *f*,
(45)$$\begin{array}{l} \frac{P_{av}R_{L}}{32\pi^{2}\gamma^{2}f^{2}P^{2}\vartheta^{2}\left(4\pi^{2}f^{2}+\mu^{2}\right)}[-4\alpha_{1}\left(4\pi^{2}f^{2}+\mu^{2}\right)+16\pi^{2}f^{2}\left(-4\pi^{2}f^{2}+\theta \vartheta +\omega^{2}\right)\\ {+4\mu^{2}\left(\omega^{2}-4\pi^{2}f^{2}\right)+\frac{3\alpha_{2}P_{av}R_{L}{\left(4\pi^{2}f^{2}+\mu^{2}\right)}^{2}}{2\pi^{2}f^{2}\vartheta^{2}}]}^{2}+\frac{2P_{av}R_{L}{\left[\nu \left(4\pi^{2}f^{2}+\mu^{2}\right)+\theta \mu \vartheta \right]}^{2}}{\gamma^{2}P^{2}\vartheta^{2}\left(4\pi^{2}f^{2}+\mu^{2}\right)}=1 \end{array}$$

### 4.2. Complex Dynamic Frequency (CDF) Method for 1/3 Subharmonic Resonance

The complex dynamic frequency (CDF) method is used to solve the 1/3 subharmonic resonance solutions of the system Equations (34) and (35). The solution of the system is set to
(46)$$\left\{ \begin{array}{l} \eta =\zeta +\bar{\zeta }+\xi +\bar{\xi },\\ \dot{\eta }=i\left(\frac{\omega_{10}}{3}+\varepsilon \omega_{11}\right)(\zeta -\bar{\zeta })+i\omega_{10}(\xi -\bar{\xi }), \end{array}\right.$$
(47)$$v=(\Gamma_{1}+i\Gamma_{2})\zeta +(\Gamma_{1}-i\Gamma_{2})\bar{\zeta }+(\Gamma_{3}+i\Gamma_{4})\xi +(\Gamma_{3}-i\Gamma_{4})\bar{\xi },$$
where $\zeta =ae^{i\omega_{10}t/3}/2$, $\bar{\zeta }=ae^{-i\omega_{10}t/3}/2$, $\xi =be^{i(\omega_{10}t+\phi )}/2$, $\bar{\xi }=be^{-i(\omega_{10}t+\phi )}/2$, and $b=-9P\gamma /(8\omega_{10}^{2})$. Differentiating Equation (46) with respect to *t*, we can obtain
(48)$$\left\{ \begin{array}{l} \dot{\eta }=\dot{\zeta }+\dot{\bar{\zeta }}+i\omega_{10}(\xi -\bar{\xi }),\\ \ddot{\eta }=i\varepsilon {\dot{\omega }}_{11}(\zeta -\bar{\zeta })+i\left(\frac{\omega_{10}}{3}+\varepsilon \omega_{11}\right)(\dot{\zeta }-\dot{\bar{\zeta }})-\omega_{10}^{2}(\xi +\bar{\xi }), \end{array}\right.,$$

Solving Equation (48) we can obtain
(49)$$\dot{\zeta }=\frac{1}{2}\left[\eta +\frac{\ddot{\eta }}{i\left(\varepsilon \omega_{11}+\frac{\omega_{10}}{3}\right)}+\frac{i\varepsilon {\dot{\omega }}_{11}\dot{\eta }}{{\left(\varepsilon \omega_{11}+\frac{\omega_{10}}{3}\right)}^{2}}+\frac{\varepsilon \omega_{10}{\dot{\omega }}_{11}\left(\xi -\bar{\xi }\right)}{{\left(\varepsilon \omega_{11}+\frac{\omega_{10}}{3}\right)}^{2}}+\frac{\omega_{10}^{2}\left(\bar{\xi }+\xi \right)}{i\left(\varepsilon \omega_{11}+\frac{\omega_{10}}{3}\right)}-i\omega_{10}\left(\xi -\bar{\xi }\right)\right],$$

By solving Equations (34), (46) and (49) simultaneously, and considering $\Omega =\omega_{10}$, we obtain
(50)$$\begin{array}{l} \left(\frac{a\omega_{10}^{2}}{18}+\varepsilon \frac{1}{3}a\omega_{10}\omega_{11}+\varepsilon^{2}\frac{1}{2}a\omega_{11}^{2}\right)\cos\left(\frac{\omega_{10}t}{3}\right)=\left(-\frac{1}{2}a\Gamma_{2}\theta -\frac{1}{6}a\nu \omega_{10}-\frac{1}{2}a\epsilon {\dot{\omega }}_{11}\right)\sin\left(\frac{\omega_{10}t}{3}\right)+\frac{3}{8}a^{2}b\alpha_{2}\cos\left(\frac{\omega_{10}t}{3}+\phi \right)\\ +\left(\frac{3a^{3}\alpha_{2}}{8}-\frac{a\alpha_{1}}{2}+\frac{3}{4}ab^{2}\alpha_{2}+\frac{1}{2}a\Gamma_{1}\theta +\frac{a\omega^{2}}{2}\right)\cos\left(\frac{\omega_{10}t}{3}\right)+\frac{3}{8}a^{2}b\alpha_{2}\cos\left(\frac{5\omega_{10}t}{3}+\phi \right)+\frac{3}{8}ab^{2}\alpha_{2}\cos\left(\frac{5\omega_{10}t}{3}+2\phi \right)\\ +\frac{3}{8}ab^{2}\alpha_{2}\cos\left(\frac{7\omega_{10}t}{3}+2\phi \right)+\frac{1}{8}b^{3}\alpha_{2}\cos(3\omega_{10}t+3\phi )+\frac{1}{8}a^{3}\alpha_{2}\cos(\omega_{10}t)+\left(-\frac{1}{2}b\Gamma_{4}\theta -\frac{1}{2}b\nu \omega_{10}\right)\sin(\omega_{10}t+\phi )\\ +\left(\frac{3}{4}a^{2}\alpha_{2}b-\frac{P\gamma }{2}+\frac{3\alpha_{2}b^{3}}{8}-\frac{\alpha_{1}b}{2}+\frac{1}{2}b\Gamma_{3}\theta +\frac{b\omega^{2}}{2}-\frac{b\omega_{10}^{2}}{2}\right)\cos(\omega_{10}t+\phi ) \end{array}.$$

Letting the coefficients corresponding to $\cos(\omega_{10}t/3)$ and $\sin(\omega_{10}t/3)$ in (50) be zero, we obtain the amplitude–frequency response relationship of the system as follows,
(51)$${\left(\frac{36\alpha_{1}-36\omega^{2}-27a^{2}\alpha_{2}-54b^{2}\alpha_{2}-36\Gamma_{1}\theta +4\omega_{10}^{2}}{27ab\alpha_{2}}\right)}^{2}+{\left(\frac{12\Gamma_{2}\theta +4\nu \omega_{10}}{9ab\alpha_{2}}\right)}^{2}=1.$$
(52)$$\tan\phi =-\frac{12\left(3\Gamma_{2}\theta +\nu \omega_{10}\right)}{36\alpha_{1}-36\omega^{2}-27a^{2}\alpha_{2}-54b^{2}\alpha_{2}-36\Gamma_{1}\theta +4\omega_{10}^{2}}$$

Balancing the other harmonic terms of the Equation (50), the dynamic frequency of the system is obtained
(53)$$\begin{array}{l} \omega_{11}=\frac{3}{80a\omega_{10}}[\left(70a^{2}b\alpha_{2}-40P\gamma +30b^{3}\alpha_{2}-40\alpha_{1}b+40b\omega^{2}-40b\omega_{10}^{2}+40b\Gamma_{3}\theta \right)\cos\left(\frac{2\omega_{10}t}{3}+\phi \right)\\ +10a^{3}\alpha_{2}\cos\left(\frac{2\omega_{10}t}{3}\right)-\left(40b\Gamma_{4}\theta +40b\nu \omega_{10}\right)\sin\left(\frac{2\omega_{10}t}{3}+\phi \right)+15ab^{2}\alpha_{2}\cos\left(2\omega_{10}t+2\phi \right)\\ +15ab^{2}\alpha_{2}\cos\left(\frac{2\omega_{10}t}{3}+2\phi \right)+30ab^{2}\alpha_{2}\cos\left(\frac{4\omega_{10}t}{3}+2\phi \right)+20a^{2}b\alpha_{2}\cos\left(\frac{4\omega_{10}t}{3}+\phi \right)\\ +b^{3}\alpha_{2}\cos\left(\frac{2\omega_{10}t}{3}+3\phi \right)+2b^{3}\alpha_{2}\cos\left(\frac{4\omega_{10}t}{3}+3\phi \right)+4b^{3}\alpha_{2}\cos\left(\frac{8\omega_{10}t}{3}+3\phi \right)+3b^{3}\alpha_{2}\cos\left(2\omega_{10}t+3\phi \right)] \end{array}.$$

Substituting Equations (46) and (47) into Equation (35), balancing corresponding harmonic terms, we obtain
(54)$$\Gamma_{1}=\frac{\omega_{10}^{2}\vartheta }{9\mu^{2}+\omega_{10}^{2}},\;\Gamma_{2}=\frac{3\mu \omega_{10}\vartheta }{9\mu^{2}+\omega_{10}^{2}},\;\Gamma_{3}=\frac{\omega_{10}^{2}\vartheta }{\mu^{2}+\omega_{10}^{2}},\;\Gamma_{4}=\frac{\mu \omega_{10}\vartheta }{\mu^{2}+\omega_{10}^{2}},$$

When 1/3 subharmonic resonance occurs in systems Equations (34) and (35), the approximate analytical solution is
(55)$$\left\{ \begin{array}{l} \eta =a\cos\left(\frac{\omega_{10}t}{3}\right)+b\cos\left(\omega_{10}t+\phi \right)\\ \dot{\eta }=-a\left(\omega_{11}+\omega_{10}\right)\sin\left(\frac{\omega_{10}t}{3}\right)-b\omega_{10}\sin(\omega_{10}t+\phi )\\ v=\frac{a\omega_{10}\vartheta }{9\mu^{2}+\omega_{10}^{2}}\left[\omega_{10}\cos\left(\frac{\omega_{10}t}{3}\right)-3\mu \sin\left(\frac{\omega_{10}t}{3}\right)\right]\\ +\frac{b\omega_{10}\vartheta }{\mu^{2}+\omega_{10}^{2}}\left[\omega_{10}\cos\left(\omega_{10}t+\phi \right)-\mu \sin\left(\omega_{10}t+\phi \right)\right] \end{array}\right.,$$

Combined with Equation (7), the actual response of the harvester when 1/3 subharmonic resonance occurs can be obtained.

## 5. Dynamic Analysis

Figure 8 shows the amplitude–frequency response diagram of the system, where Figure 8a is the primary resonance. The purple line is the amplitude–frequency relationship obtained with the multi-scale method. The multi-scale method [31,32] has been widely used in the research of energy harvesters, so the iterative process of the multi-scale method is not listed in this paper. The multi-scale method effectively solves weakly nonlinear vibration problems, but accurate results cannot be obtained for strongly nonlinear vibration problems. It can be seen from Figure 8a that for the object studied in this paper, the results obtained using the multi-scale method differ significantly from the numerical solutions. The results obtained using the CDF method are in better agreement with the numerical solution. Therefore, it is feasible to use the CDF method for the theoretical analysis in this paper. As shown from Figure 8a when the excitation frequency is between 8.5 and 17.4 Hz, one frequency corresponds to multiple amplitudes in the system, one unstable solution, and two stable solutions. The larger values represent the inter-well motion in the stable solution, and the smaller values represent the intra-well motion. The trajectories of the response amplitudes during forward and reverse sweep are indicated by arrows ① and ②, respectively. The steady-state response at the former excitation frequency during the sweep is the initial condition for the vibration at the latter excitation frequency. It is easily obtained that the steady-state motion amplitude of the system depends on the initial condition in the frequency range of single-frequency multi-amplitude. As shown in Figure 4, the system has two centers and a saddle point. When the initial conditions are different, some orbits in the system surround the center, and the other orbits are around the three fixed points, which leads to the results in Figure 8a. The amplitude–frequency response of the corresponding linear system is shown in Figure 8b. It can be seen from the figure that the linear system does not have a single-frequency multi-amplitude phenomenon, and the forward and reverse sweep results are the same. In Figure 8a, the frequency range of amplitude greater than 4 mm is 0~17.4 and 0~8.5 Hz for the forward sweep and reverse sweep, respectively. In Figure 8b, the frequency range of amplitude greater than 4 mm is 23.5~25.8 Hz. It can be seen that the introduction of strong nonlinearity in the harvester not only reduces the resonant frequency but also expands the effective frequency range by at least 270%.

Figure 8c gives the amplitude–frequency response relationship of the nonlinear system when 1/3 subharmonic resonance occurs. It can be seen from the figure that only in a specific frequency range, i.e., 10.1~17.9 Hz, the system is likely to undergo 1/3 subharmonic resonance. Comparing Figure 8a,c, it is found that the frequency range of the single-frequency multi-amplitude occurring in the primary resonance is roughly the same as that of the 1/3 subharmonic resonance. In this frequency range, due to different initial conditions of the system, three different steady-state responses can be generated: large-amplitude vibration between wells, small-amplitude vibration in wells, and 1/3 subharmonic vibration.

Figure 9 is the relationship between the average power and the excitation frequency of the nonlinear system, where Figure 9a is the primary resonance and can be obtained from Equation (45), Figure 9b is 1/3 subharmonic resonance, similar to the primary resonance, which can be obtained by deformation of Equation (51). In the primary resonance, the maximum power is 1.8 mW when the excitation frequency is 17.3 Hz, and the minimum power is less than 0.01 mW when the excitation frequency is 15~17.3 Hz. However, according to Huguet et al. [16], the inter-well motion is less robust in the range of 8.5 to 17.3 Hz, so the maximum power obtainable is 0.7 mW. When 1/3 subharmonic resonance occurs, the maximum power obtained is 0.25 mW, which is 36% of the maximum power of the primary resonance and exceeds 2500% of the minimum power of the primary resonance. Although the power of the subharmonic resonance is smaller than the power of the inter-well motion of the primary resonance, it is much larger than the intra-well motion of the primary resonance. Therefore, the 1/3 subharmonic resonance expands the band of the nonlinear energy harvester by 92% so that the energy harvester can obtain more than 0.1 mW power in the frequency range of 18 Hz. Meanwhile, Figure 8 and Figure 9 prove that the CDF method can predict the primary resonance and 1/3 subharmonic resonance behavior.

Figure 10 shows the spectrum of the system at different excitation frequencies when the 1/3 subharmonic resonance occurs. It can be found that the response contains two frequency components, indicating that when 1/3 subharmonic resonance occurs, the primary harmonic and the 1/3 subharmonic coexist. When the excitation frequency increases, the primary harmonic component of the response becomes smaller and smaller, and the 1/3 subharmonic component becomes larger and larger as a whole. Combined with Figure 8c, the same conclusion can be drawn by observing that the primary harmonic coefficient *a* increases and the 1/3 subharmonic coefficient *b* decreases as the excitation frequency increases in Equation (55).

Figure 11 illustrates the components of the time-domain diagram when the system undergoes 1/3 subharmonic resonance. Figure 11c,f shows the time-domain diagrams of the system’s displacement when 1/3 subharmonic resonance occurs at the excitation frequency *f* = 14 Hz and *f* = 17.5 Hz, respectively. When the system undergoes 1/3 subharmonic resonance, the response contains the primary harmonic and 1/3 harmonic. Figure 11c is superimposed from the results of Figure 11a,b, where Figure 11a is the primary harmonic and Figure 11b is the 1/3 harmonic. Since the period of the 1/3 harmonic is three times that of the primary harmonic, the superposition results in the same period as the 1/3 harmonic period. Similarly, Figure 11f is a superposition of the results of Figure 11d,e. Comparing Figure 11a,b,d,e, it can be seen that the higher the excitation frequency, the smaller the primary harmonic component and the larger the 1/3 harmonic component. The primary harmonic can be regarded as a small perturbation applied to the 1/3 harmonic. The smaller the perturbation, the more stable the superposition result.

The conditions for 1/3 subharmonic resonance of the system are discussed below. Equation (51) is rewritten to
(56)$$\begin{array}{l} 729\alpha_{2}^{2}{\tilde{a}}^{2}-\left[729b^{2}\alpha_{2}^{2}+54\alpha_{2}\left(36\alpha_{1}-36\omega^{2}-54b^{2}\alpha_{2}-36\Gamma_{1}\theta +4\omega_{10}^{2}\right)\right]\tilde{a}\\ +{\left(36\alpha_{1}-36\omega^{2}-54b^{2}\alpha_{2}-36\Gamma_{1}\theta +4\omega_{10}^{2}\right)}^{2}+{\left(36\Gamma_{2}\theta +12\nu \omega_{10}\right)}^{2}=0 \end{array}$$
where $\tilde{a}=a^{2}$. According to Weda’ s theorem, the condition for the existence of positive real roots is
(57)$$\left\{ \begin{array}{l} \left(72\omega^{2}-72\alpha_{1}+72\Gamma_{1}\theta -8\omega_{10}^{2}+81\alpha_{2}b^{2}\right)\alpha_{2}<0,\\ -b^{2}\alpha_{2}\left[2551.5b^{2}\alpha_{2}-54\left(36\alpha_{1}-36\Gamma_{1}\theta -36\omega^{2}+4\omega_{10}^{2}\right)\right]>288{\left(3\Gamma_{2}\theta +\nu \omega_{10}^{2}\right)}^{2}, \end{array}\right.$$

It can be seen from Equation (57) that the conditions for the 1/3 subharmonic resonance of the nonlinear system are related to excitation frequency, excitation amplitude, linear stiffness coefficient, and nonlinear stiffness coefficient. Figure 12 is derived from Equation (57) and shows the range of the excitation amplitudes and frequencies that can occur for the 1/3 subharmonic resonance with other parameters held constant. When *p* < 1.5 m/s^2^, the 1/3 subharmonic resonance does not occur at any excitation frequency; when *f* < 5.5 Hz, the 1/3 subharmonic resonance does not occur at any excitation amplitude. In the region where the 1/3 subharmonic resonance can occur in the system, the corresponding excitation frequency range increases with excitation amplitude. Similarly, as the excitation frequency increases, the corresponding excitation amplitude range increases. Therefore, for a piezoelectric energy harvester whose parameters have been determined, the method can be used to grasp the external excitation conditions for its generation of the 1/3 subharmonic resonance; if a piezoelectric energy harvester is to be designed to broaden the bandwidth using the 1/3 harmonic resonance under specific environmental conditions, the method can be used to invert the parameters of the harvester and provide theoretical guidance for the design of the harvester.

Figure 13 is the diagram of the relationship between the response displacement amplitude and the excitation amplitude of the system (if not specified, it refers to the case of the primary resonance). When the excitation amplitude is less than 5.9 m/s^2^, the system has only a small vibration; when the excitation amplitude is 5.9~36.7 m/s^2^, the system has a single excitation amplitude corresponding to multiple response amplitudes, one unstable solution, and two stable solutions; and when the excitation amplitude is greater than 36.7 m/s^2^, the system can maintain large vibration. The trajectories of response amplitudes during forward and reverse sweep are shown by arrows ③ and ④, respectively. When sweeping, the steady-state response under the previous excitation amplitude is the initial condition for the vibration at the next excitation amplitude. It can be easily seen that the steady-state motion of the system depends on the initial condition in the multi-value excitation amplitude region. This is similar to Figure 8a, where there is a jump during sweeping. For the range of excitation amplitudes that can produce multi-values, we cannot guarantee that the system always vibrates with large amplitudes. If the system performs small amplitude vibration, the efficiency of the energy harvester is greatly reduced. We are interested in considering the effect of 1/3 subharmonic motion.

Figure 14 shows the relationship between response amplitude and excitation amplitude under different excitation frequencies obtained using the CDF method. When the excitation amplitude is in the range of 0 to 10 m/s^2^ and the excitation frequency is 4, 10, and 16 Hz, the excitation amplitude range of the system with large vibration is 0.7~10, 3.2~10, and 7.7~10 m/s^2^, respectively; the range of excitation amplitude that can ensure the system vibrates greatly is 1.5~10 m/s^2^, 0, and 0 respectively. High-frequency excitation can obtain a large amplitude response, thus obtaining a higher output power. However, the excitation amplitude range of multi-values is wide, and only small vibration in the multi-values may be obtained, so as to obtain the smaller output. At this time, the low-frequency excitation can obtain a larger response amplitude, but this limits the bandwidth of the system. If the 1/3 subharmonic motion is considered, when the excitation frequencies are 4, 10, and 16 Hz, the excitation amplitude ranges of 1/3 subharmonic motion in the system are 0, 3.2~8.7, and 7.2~32.7 m/s^2^. The excitation amplitude range of 1/3 subharmonic motion is basically in the range of multi-valued excitation amplitude generated by primary resonance, and it enlarges with the increase in excitation frequency. Although the amplitude of 1/3 subharmonic resonance is smaller than that of the large amplitude motion of the primary resonance, it is larger than that of the small amplitude of the primary resonance. If 1/3 subharmonic motion is considered instead of the small amplitude motion of the primary resonance, the harvester can obtain more energy. For *f* = 4 Hz, the system has no 1/3 harmonic resonance, consistent with Figure 12. Combined with the previous analysis, when *p* > 1.5 m/s^2^, *f* > 5.5 Hz, 1/3 subharmonic resonance occurs in the system, and when *p* is larger, the frequency range of 1/3 subharmonic resonance is wider. At this point, the 1/3 subharmonic resonance can replace intra-well motion, so that the harvester can obtain higher energy output in the broader frequency range. In practical application, the CDF method can provide theoretical guidance for the parameter design of the harvester to obtain higher power output in the environment of low-frequency excitation and small excitation amplitude.

## 6. Conclusions

In this paper, the influence of nonlinear characteristics on the bandwidth of a piezoelectric vibration energy harvester is studied using the CDF method. Firstly, we modified the model, considered the influence of torque on the bending of the cantilever beam when modeling with the Hamiltonian principle, and compared it with the situation without considering the torque. It is found that torque greatly influences the equilibrium point and piezoelectric output of the harvester. In the range of bistable parameters, with the increase in permanent magnet spacing, the influence of torque also increases. Therefore, the influence of torque should be considered in the theoretical modeling. Secondly, the static analysis of the system is carried out to obtain the static bifurcation behavior and determine the parameter conditions for the bistable state of the harvester. Then, the approximate analytical solution and amplitude–frequency relationship of the primary resonance and 1/3 subharmonic resonance of the system are obtained using the CDF method. Finally, the CDF method is used to analyze the dynamic behavior of the energy harvester. The introduction of nonlinearity makes the harvester produce a bistable state and increases the bandwidth of primary resonance. When the 1/3 subharmonic vibration occurs in the harvester, it mainly contains the primary harmonic and the 1/3 subharmonic, and with the increase in the excitation frequency, the larger the proportion of the subharmonic. At the same time, it is found that the frequency range of 1/3 subharmonic vibration is within the frequency range of multi-values generated by the primary resonance, and the subharmonic motion’s amplitude is greater than that of the intra-well motion. Similarly, 1/3 subharmonic resonance also increases the excitation amplitude’s bandwidth of the harvester at a specific excitation frequency. Therefore, to a certain extent, the subharmonic motion can compensate for the low energy collection efficiency caused by the low robustness of inter-well motion. Making full use of the nonlinear characteristics of a system can effectively broaden the bandwidth of the piezoelectric vibration energy harvester and improve the efficiency of obtaining environmental energy. The theoretical method (CDF) can accurately predict the dynamic behavior of this kind of strongly nonlinear vibration energy harvester and provide guidance for its theoretical research.

In the next step, the parameters of the harvester model are optimized by considering the effects of residual stress, temperature, damping, and the dimensions of the multilayer beam. On this basis, the following studies will be carried out.

① An optimal load resistance exists in the system. A more significant power can be output under the same conditions when the optimal load resistance is obtained. However, the variation of the load resistance may affect the dynamic behavior of the system. Determining the optimal load by integrating the mechanical system and the circuit system remains to be studied.

② Subharmonic resonance occurs only within a specific range related to the excitation frequency, excitation amplitude, linear stiffness coefficient, and nonlinear stiffness coefficient. By adjusting these parameters, the subharmonic resonance can maximize the bandwidth of the harvester.

## Figures and Tables

**Figure 1 micromachines-12-01301-f001:**
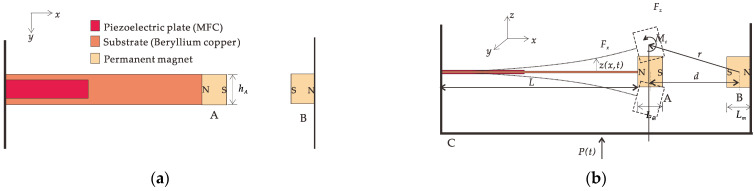
Nonlinear piezoelectric cantilever beam model: (**a**) front view; (**b**) vertical view.

**Figure 2 micromachines-12-01301-f002:**
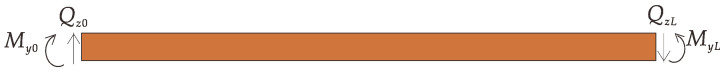
Schematic diagram of piezoelectric cantilever beam boundary force.

**Figure 3 micromachines-12-01301-f003:**
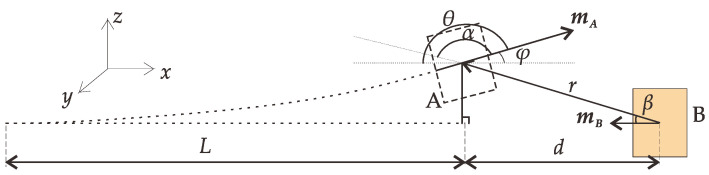
Geometric diagram.

**Figure 4 micromachines-12-01301-f004:**
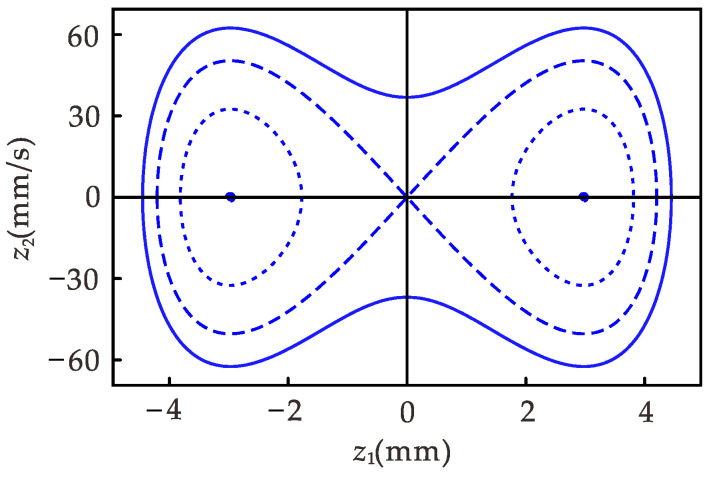
Contour diagram of the energy function.

**Figure 5 micromachines-12-01301-f005:**
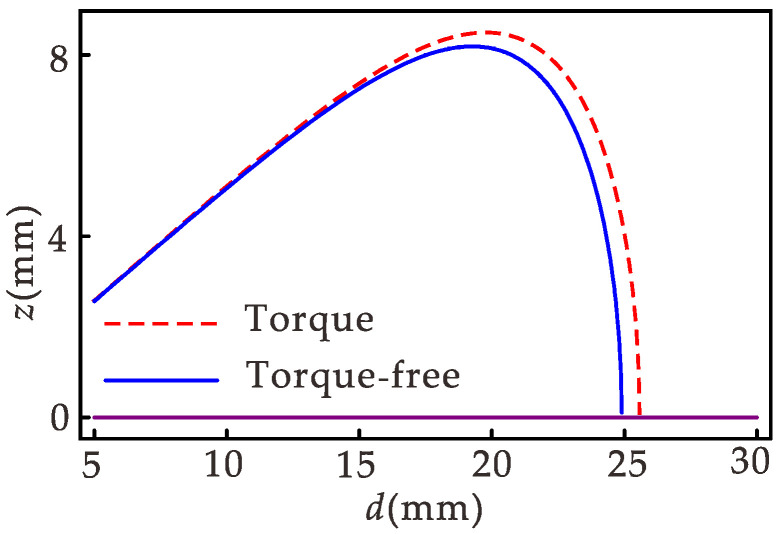
Equilibrium point bifurcation diagram.

**Figure 6 micromachines-12-01301-f006:**
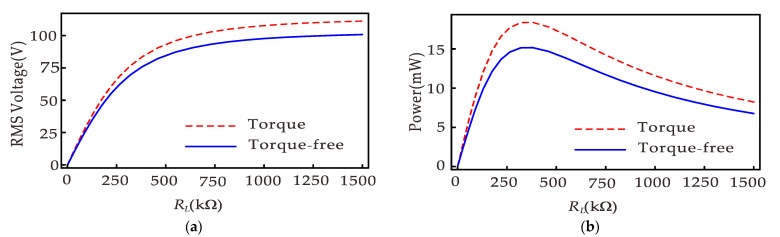
Piezoelectric output versus load resistance: (**a**) RMS voltage; (**b**) output power.

**Figure 7 micromachines-12-01301-f007:**
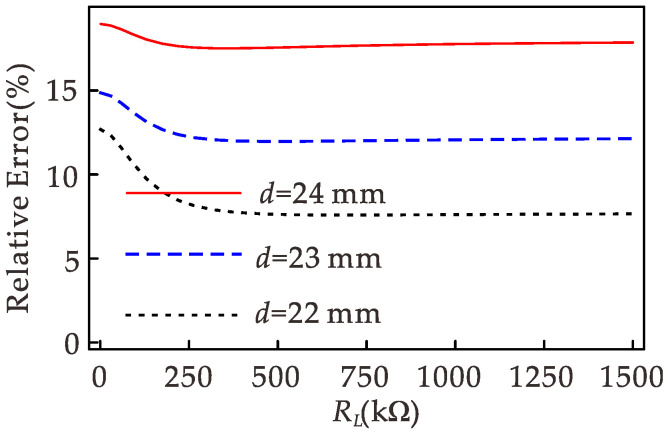
The relative error of the RMS power versus the load resistance when the torque is not considered.

**Figure 8 micromachines-12-01301-f008:**
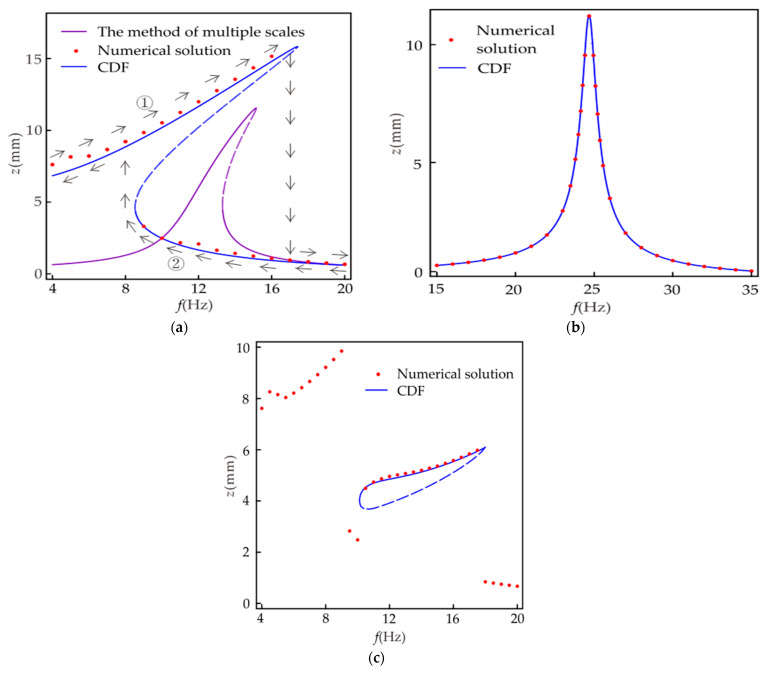
Displacement versus frequency with *p* = 9 m/s^2^, *d* = 24 mm: (**a**) nonlinear system (primary resonance); (**b**) linear system (primary resonance); (**c**) nonlinear system (1/3 subharmonic resonance).

**Figure 9 micromachines-12-01301-f009:**
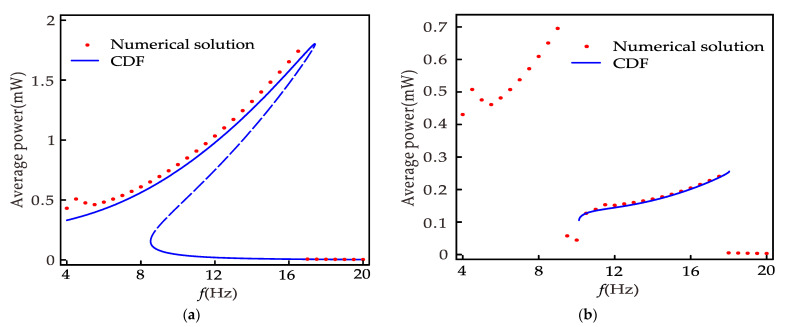
Average power versus frequency with *p* = 9 m/s^2^, *d* = 24 mm, *R_L_* = 1000 kΩ: (**a**) nonlinear system (primary resonance); (**b**) nonlinear system (1/3 subharmonic resonance).

**Figure 10 micromachines-12-01301-f010:**
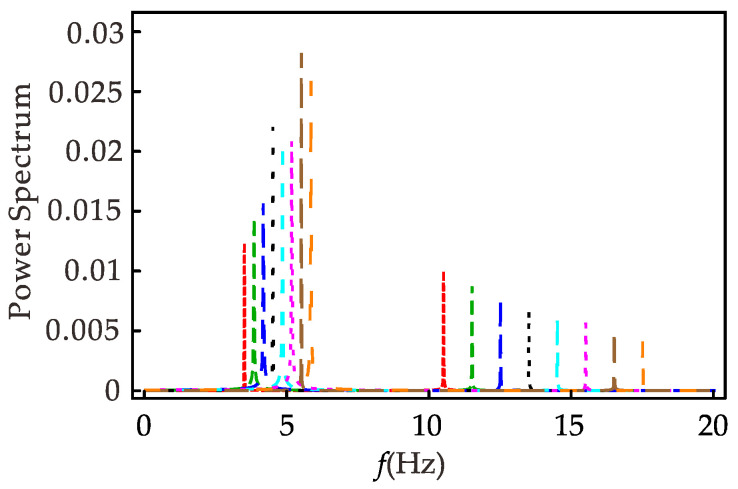
Spectrum diagram of 1/3 subharmonic resonance at different excitation frequencies.

**Figure 11 micromachines-12-01301-f011:**
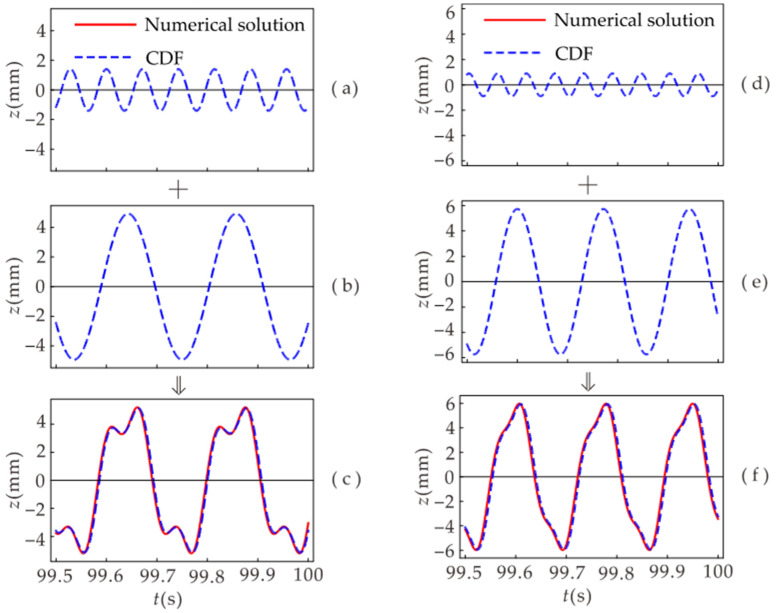
Displacement time-domain diagram: displacement time-domain diagrams when the excitation frequency *f* = 14 Hz and *f* = 17.5 Hz, respectively. (**a**,**d**) represent primary harmonic; (**b**,**e**) represent 1/3 harmonic. (**c**,**f**) represent the 1/3 subharmonic resonance.

**Figure 12 micromachines-12-01301-f012:**
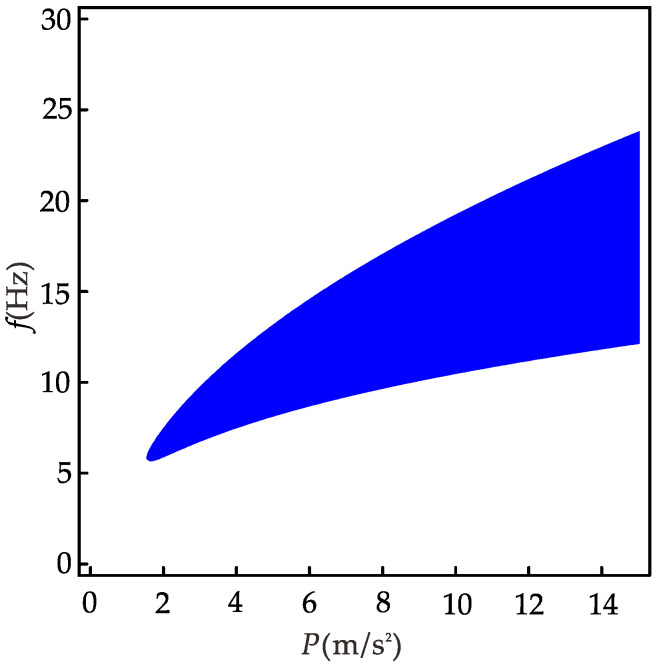
The region of 1/3 subharmonic resonance of the nonlinear system.

**Figure 13 micromachines-12-01301-f013:**
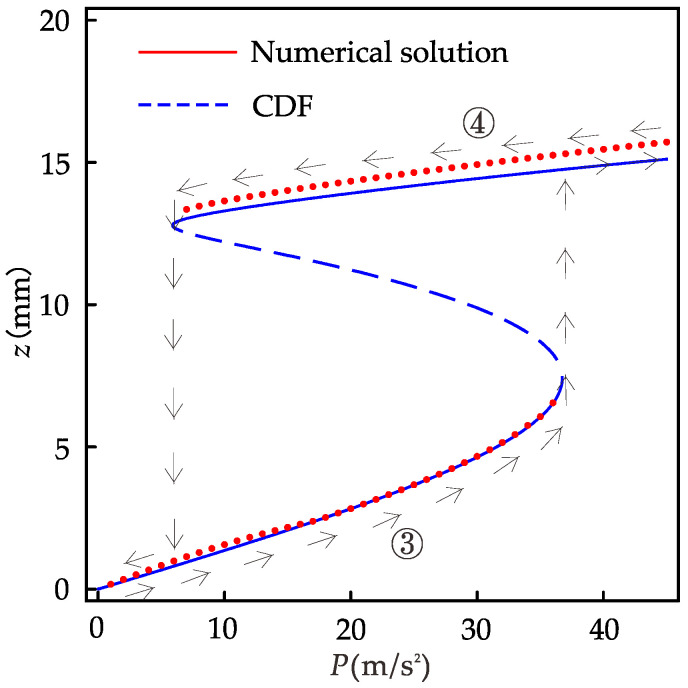
Displacement amplitude versus frequency with *f*=14 Hz, *d*=24 mm.

**Figure 14 micromachines-12-01301-f014:**
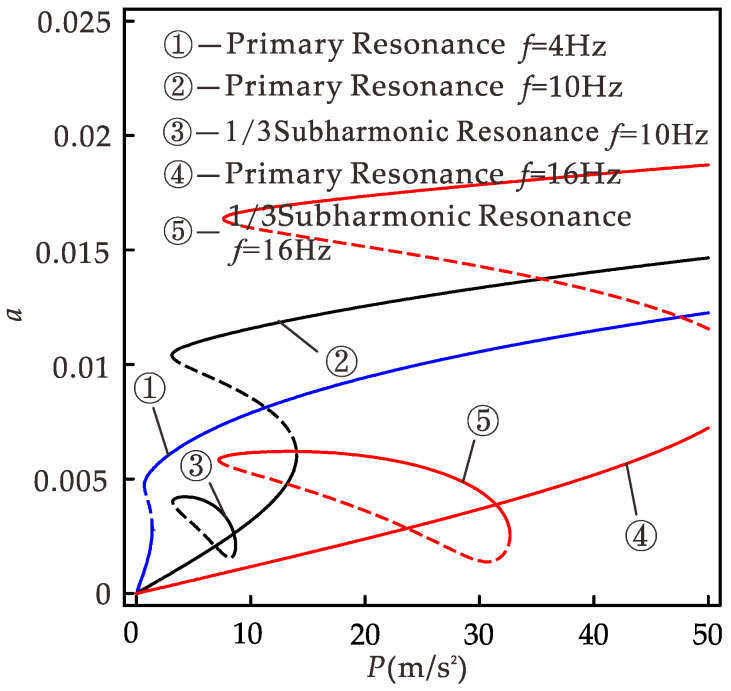
Amplitude versus excitation amplitude with different excitation frequencies.

**Table 1 micromachines-12-01301-t001:** Physical parameter values of the piezoelectric cantilever beam vibration system.

	Parameter	Symbol	Value
Substrate properties	Length	L	85 mm
	Width	b	15 mm
	Thickness	$h_{s}$	0.6 mm
	Density	$\rho_{s}$	8300 kg/m^3^
	Young’s modulus	$E_{s}$	130 GPa
Piezoelectric plate properties	Length	$L_{p}$	37 mm
	Width	$b_{p}$	10 mm
	Thickness	$h_{p}$	0.3 mm
	Density	$\rho_{p}$	5440 kg/m^3^
	Young’s modulus	$E_{p}$	30,336 GPa
	Piezoelectric constant	$d_{31}$	−170 pC/N
Permanent magnet properties	Length	$L_{m}$	10 mm
	Height	$h_{A},\;h_{B}$	15 mm
	Thickness	$l_{A},\;l_{B}$	5 mm
	Density	$\rho_{A},\;\rho_{B}$	7500 kg/m^3^
	Residual flux density	$B_{r}$	1.25 T
	Vacuum permeability	$\mu_{0}$	$4\pi \times 10^{-7}\;N/A^{2}$

## Data Availability

Not applicable.

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
