# Peer review of "Theoretical Study on Widening Bandwidth of Piezoelectric Vibration Energy Harvester with Nonlinear Characteristics"

_micromachines, 2021, doi:10.3390/mi12111301_

Round 1
Reviewer 1 Report
This manuscript presents a theoretical modeling on widening bandwidth of piezoelectric vibration energy harvester considering nonlinear parameters. This modeling could be used in design of piezoelectric vibration energy harvester to increase the power output. The results of the theoretical modeling were compared with those of complex dynamic frequency (CDF) method. However, this manuscript can be improved considering the following comments:
1.-Introduction section should include recent references (2019-2021) about analytical modeling of piezoelectric vibration energy harvesters.
2.-Authors should add the main advantages of the proposed modeling in comparison with other reported in the literature.
3.-Second section should mention the main assumptions (e.g., Bernoulli beam, size of beam, material type, multilayers, residual stress, damping sources) used in the theoretical model. What is the minimum thickness of the cantilever beam considered in the theoretical modeling?
4.-May the proposed modeling consider the influence of multilayers cantilever beams?
5.-What piezoelectric material type is considered in the theoretical modeling? Authors must include more discussions about the piezoelectric material type that can be considered in the proposed modeling.
6.-Which are the main limitations of the proposed modeling?
7.-Authors should add discussions about the effect of residual stress, temperature, damping, and size of multilayer beam on the results of theoretical modeling.
8.-Authors should compare the results of analytical model with respect to experimental data.
9.-Description of all variables or parameters used in equations must be checked. For instance, parameters of equations (9) and (11).
10.-Captions of Figures 6 and 10 must be improved.
11.-Authors should include more discussion about results shown in Figures 11-15.
12.-Conclusion section should be improved considering the modifications of the manuscript. Which are the future research works?
Reviewer 2 Report
Article Review
Q.Zhang, Y.Yang and W.Wang, Theoretical Study on Widening Bandwidth of Piezoelectric Vibration Energy Harvester by Nonlinear Characteristics. Micromachines, 2021.
In this work, researchers focused on widening operation bandwidth of the proposed piezoelectric energy harvester by introducing nonlinearity to the system. Nonlinearity was also added into the system to help the piezoelectric vibration energy harvester collect more energy over a wider frequency range, which can cause the harvester to generate several steady states as well as to deflect the frequency response curve. However, researchers stated that it is far easier for the harvester to sustain intra-well motion rather than inter-well motion, which has a significant impact on the collector's efficiency. In the theoretical modeling, the effect of torque between permanent magnets on the bending of the piezoelectric cantilever beam was considered. The complex dynamic frequency (CDF) method was used to obtain approximate analytical solutions of the primary resonance and subharmonic resonance of the harvester. The results reveal that the torque has a significant impact on the harvester's equilibrium point and piezoelectric output. Researchers claim that bistable nonlinear energy harvester's effective frequency bandwidth is 270 percent broader than the linear harvesters, and the 1 / 3 subharmonic resonance broadens the frequency band by 92 percent further, allowing the energy harvester to generate more than 0.1 mW power in the 18 Hz frequency range.
Researchers have started the article with a good introduction explaining the significance of the problem, supporting their claim using numerous academic sources. They explained why they selected a cantilever structure for their application. Modeling of the device was explained analytically using scientific principles and every iteration of those calculations were given explicitly in the article. At the end of most of the chapters, results of the analytical formulations were given in plots which is an indicator of a high-quality presentation. In the dynamic analysis part, CDF values and numerical calculations were compared clearly. The problem of having a broader resonance frequency bandwidth for energy harvester operation was a well-defined problem which is commonly encountered in energy harvester development. The authors approached the problem by developing a bistable device. All the results were given appropriately in plots which is easier for the audience to interpret. They are also justified with numerous analytical formulas. The increase in the operation bandwidth reported in the article is a significant result which can be also adopted by other researchers building their own energy harvesters. Researchers used an appropriate level of English in this article.
However, the information given about the device itself was not enough. The researchers gave information about design specifications to give a broad perspective about what type of a device will be built. Researchers should have given information about the possible fabrication sequence of the device. How are the magnets are going to be placed? Which materials are going to be used? Young’s Modulus and piezoelectric coefficients of those materials are given in the article however, the audience don’t know the name of these materials yet.
3D representation of the cantilever at the beginning of the article as Fig 1. would be helpful for the audience to understand the concept.
Another question is the significance of the CDF method here. Researchers don’t have a clear explanation of why they are using CDF in the simulations.
At line 83, there are two articles written by the researchers, citations 8 and 9, which doesn’t have any significance in the article. Also, citation 17 at line 119 is also another article from the researcher. The article is cited for describing the CDF method. Showing his/her own article as a standing point for the one of the most important part of this research doesn’t seems right. There should be at least 2-3 other supportive articles describing the usefulness of the CDF method.
At line 195, the opening sentence of the paragraph contains “many researchers provide…”. Please provide references.
In overall, this article has many significant results which may contribute to the scientific area of energy harvesters. It’s a good, well written article with an academic level of English, good graphics, and comparisons between different analytical methods are given. To make it even better, researchers can make some minor changes such as supporting their claim more strongly in terms of CDF selection and giving some more details in the device design part.
Round 2
Reviewer 1 Report
This revised manuscript has been improved considering the reviewer's comments. This manuscript is suitable for publication in Micromachines.